

# CTLA-4 polymorphisms associate with breast cancer susceptibility in Asians: a meta-analysis

Zhiming Dai[1,2,*], Tian Tian[1,*], Meng Wang[1,*], Xinghan Liu[1], Shuai Lin[1], Pengtao Yang[1], Kang Liu[1], Yi Zheng[1], Peng Xu[1], Meng Liu[1], Xuewen Yang[1] and Zhijun Dai[1]

[1] Department of Oncology, Second Affiliated Hospital of Xi'an Jiaotong University, Xi'an, China
[2] Department of Anesthesia, Second Affiliated Hospital of Xi'an Jiaotong University, Xi'an, China
[*] These authors contributed equally to this work.

Corresponding author
Zhijun Dai, dzj0911@xjtu.edu.cn,
dzj0911@126.com

## ABSTRACT

Previous studies have investigated the association between cytotoxic T-lymphocyte antigen-4 (*CTLA-4*) polymorphisms and breast cancer susceptibility, but the results remained inconsistent. Therefore, we evaluated the relationship between four common *CTLA-4* polymorphisms and breast cancer risk by a meta-analysis, aiming to derive a comprehensive and precise conclusion. We searched EMBASE, Pubmed, Web of Science, CNKI, and Wanfang databases until July 18th, 2016. Finally, ten eligible studies involving 4,544 breast cancer patients and 4,515 cancer-free controls were included; all these studies were from Asia. Odds ratio (OR) and 95% confidence interval (CI) were used to evaluate the breast cancer risk in five genetic models. The results indicated that the *CTLA-4* +49A>G (rs231775) polymorphism had a significant association with decreased breast cancer risk in allelic, homozygous, dominant and recessive models. Also, the +6230G>A (rs3087243) polymorphism reduced breast cancer risk especially in the Chinese population under homozygous and recessive models. In contrast, the −1661A>G (rs4553808) polymorphism increased breast cancer risk in allelic, heterozygous and dominant models, whereas −1722 T>C (rs733618) did not relate to breast cancer risk. In conclusion, *CTLA-4* polymorphisms significantly associate with breast cancer susceptibility in Asian populations, and different gene loci may have different effects on breast cancer development. Further large-scale studies including multi-racial populations are required to confirm our findings.

## INTRODUCTION

Breast cancer has been the most common type of cancer and the main cause of cancer death among women in the world, which was estimated to have 1.7 million new cases in 2012 (*Torre et al., 2015*). Breast cancer is an extremely heterogeneous disease in the clinic and the potential molecular mechanism of carcinogenesis has not been clearly understood so far. In recent years, inherited factors were identified to play a critical role in the development of breast cancer (*Reeves et al., 2012*).

Researches on the field of tumour immunology found that the immune system can influence tumour occurence during the period of elimination, equilibrium and escape (*Dunn, Old & Schreiber, 2004*). Cytotoxic T-lymphocyte antigen-4 (CTLA-4), which was also designated as CD152, expressed mainly on activated T cells. As an immunosuppressive cytokine, it can inhibit T-lymphocyte proliferation and activation (*Sun et al., 2008*). Numerous researches have demonstrated that blockage of CTLA-4 function can improve antitumor immunity (*Leach, Krummel & Allison, 1996*; *Ribas et al., 2004*; *Vandenborre et al., 1999*). This indicates CTLA-4 may exert positive effects on carcinogenesis. The human *CTLA-4* gene, which is located in human chromosome 2q33, is one of the most important genes involved in immune responses to a variety of antigens (*Walunas et al., 2011*). *CTLA-4* gene comprises four exons and has several important polymorphisms in the entire region, including the +49G>A (rs231775) in exon 1 (*Donner et al., 1997*), the +6230G>A (rs3087243) in 3′-untranslated region (*Hughes, 2006*), the −1661A>G (rs4553808) and −1722 T>C (rs733618) in the promoter region (*Johnson et al., 2001*), which are the most commonly studied single nucleotide polymorphisms (SNPs). These SNPs are important because they can alter the host immune response by affecting the transcription of *CTLA-4* gene, the expression of CTLA-4 protein, and the interaction of CTLA-4 and CD80 ligand (*Anjos et al., 2002*; *Sun et al., 2008*; *Wang et al., 2002*).

Numerous investigations have demonstrated that *CTLA-4* genetic polymorphisms may have association with human breast cancer susceptibility (*Erfani et al., 2006*; *Ghaderi et al., 2004*; *Li et al., 2012*; *Li et al., 2008*; *Minhas et al., 2014*; *Sun et al., 2008*; *Wang et al., 2007*; *Zhifu et al., 2015*; *Kong, 2010*). The results showed that some of the polymorphisms such as rs733618 and rs4553808 may increase the breast cancer risk while other polymorphisms such as rs231775 and rs3087243 may reduce the risk of breast cancer. Considering that a single study does not have enough power to detect the overall effects, we conducted a meta-analysis which is a statistical analysis of the data from some collection of studies in order to synthesize the results to obtain a more reliable evaluation of the relationship between the four common SNPs in *CTLA-4* gene and breast cancer susceptibility.

## MATERIALS AND METHODS

Our meta-analysis was conducted according to the Preferred Reporting Items for Systematic Reviews and Meta-Analyses (PRISMA) guidelines (*Moher et al., 2010*).

### Search strategy

We searched the databases of EMBASE, PubMed, Web of Science, Wanfang, as well as Chinese National Knowledge Infrastructure (CNKI) to identify all the relevant articles up to July 18th, 2016. Keywords for search were: "CTLA-4 or Cytotoxic T-lymphocyte antigen 4 or CD152," "polymorphism or variation or SNP or rs231775 or rs308724 or rs4553808 or rs733618," and "breast cancer." The eligible article must be published in English or Chinese. References in retrieved articles were also searched manually.

### Criteria for selection

All studies selected for further meta-analysis must conform to the included criteria: (1) case-control study conducted in human and investigated the association of SNPs in

*CTLA-4* with breast cancer susceptibility; (2) All the breast cancer patients were diagnosed by pathology or histology; (3) detailed data of the allele and genotype distributions are available; (4) the controls were cancer-free individuals. In addition, articles meet the following criteria were excluded: (1) articles that were reviews, conference abstracts, or repeat publications; (2) study design were based on family; (3) studies with no control groups. The quality of each included study was assessed by the Newcastle-Ottawa Scale for case-control studies (*Wells et al., 2014*).

### Data extraction
According to the selection criteria, two authors (Zhiming Dai and Tian Tian) reviewed the literature independently and extracted the raw data and information from each eligible study including: first author, publication year, country of origin, racial ancestry, source of control, genotype method, total number of case and control, allele frequency and genotype distribution in case and control, and *P* value of HWE in control. Any discrepancy was discussed between authors and refereed by Zhijun Dai to reach a consensus.

### Statistical analysis
For each study, ORs and 95% CIs were computed to estimate the breast cancer risk associated with *CTLA-4* polymorphisms. Pooled ORs were calculated under the following genetic models: allele comparison of B *vs.* A, homozygote of BB *vs.* AA, heterozygote of AB *vs.* AA, dominant model of (BB + AB) *vs.* AA and recessive model of BB *vs.* (AA + AB). Heterogeneity among studies were estimated by $I^2$ test and chi$^2$-based Q statistic and significance was considered at $I^2 > 50\%$ (*Higgins & Thompson, 2002*). We adopted the random-effects model to analyze the combined ORs if $I^2$ value was greater than 50%. Otherwise, a fixed-effects model should be exerted (*Petitti, 2001*). We carried out subgroup analysis to estimate the specific effects of ethnicity and source of control. We also conducted a sensitivity analysis to assess the consistency and stability of our meta-analysis by omitting individual study in turn. Additionally, Begg's funnel plot and Egger's test were used to assess publication bias, and significance was identified as $P < 0.05$ (*Begg & Mazumdar, 1994*; *Egger et al., 1997*). All the statistical analyses were implemented with the Review Manager (Version 5.3; Cochrane Collaboration, London, UK) and STATA software (Version 12.0; Stata Corp, College Station, TX, USA).

## RESULTS
### Characteristics of included studies
Finally, nine articles comprising 10 studies investigating *CTLA-4* +49A>G (rs231775) and/or +6230G>A (rs3087243) and/or −1722 T>C (rs733618) and/or −1661A/G (rs4553808) polymorphisms were identified for further analysis (Fig. 1). Table 1 presented the characteristics of selected studies. Of the 10 studies, seven were from China, two were from Iran, and one was from India. Additionally, seven studies were based on population and three based on hospitals. Moreover, genotype distributions in the control group of each included studies complied with Hardy-Weinberg equilibriums (HWE) ($P > 0.05$) except for one study for one SNP (*Erfani et al., 2006*). The detailed data of the allele frequency and genotype distribution as well as HWE from each study were shown in Table 2.

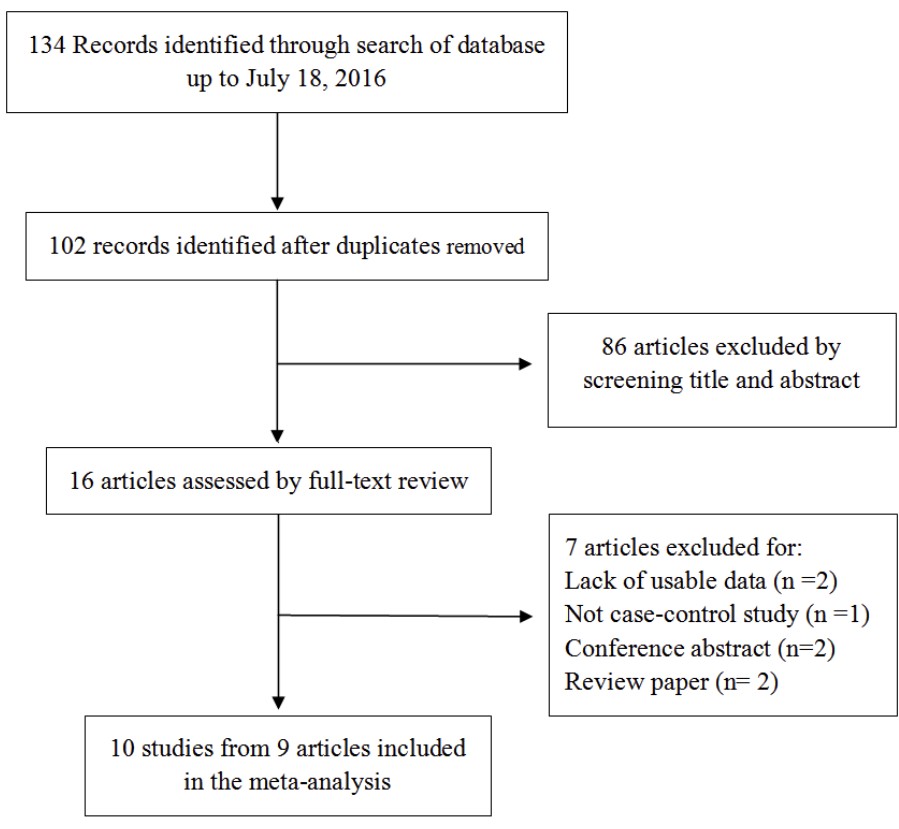

**Figure 1  Flow chart of the studies selection.**

**Table 1  Characteristics of the studies included in the meta-analysis.**

| First author | Year | Country | Ethnicity | Genotyping medthod | Source of control | Total sample size (case/control) | SNP |
|---|---|---|---|---|---|---|---|
| Yu | 2015 | China | Chinese | PCR-RFLP | PB | 376/366 | 1;2;3;4 |
| Minhas | 2014 | India | Indian | PCR-RFLP | PB | 250/250 | 1 |
| Li D | 2012 | China | Chinese | PCR-RFLP | PB | 581/566 | 1;2;3;4 |
| Kong | 2010 | China | Chinese | PCR-RFLP | HB | 315/322 | 4 |
| Sun | 2008a | China | Chinese | PCR-RFLP | PB | 1060/1070 | 1 |
| Sun | 2008b | China | Chinese | PCR-RFLP | PB | 1037/1070 | 1 |
| Li H | 2008 | China | Chinese | PCR-RFLP | HB | 328/327 | 2;3 |
| Wang | 2007 | China | Chinese | PCR-RFLP | PB | 117/148 | 1;2;4 |
| Erfani | 2006 | Iran | Iranian | PCR-CTPP | PB | 283/245 | 3;4 |
| Ghaderi | 2004 | Iran | Iranian | PCR-RFLP | HB | 197/151 | 1 |

Notes.
PCR, polymerase chain reaction; RFLP, restriction fragment length polymorphism; CTPP, confronting two pairs primers; PB, population based; HB, hospital based; SNP, single-nucleotide polymorphism.
SNP No.1, +49A>G (rs231775); 2, +6230G>A (rs3087243); 3, −1722T>C (rs733618); 4, −1661A>G (rs4553808).

**Table 2  Genotype distributions and allele frequencies of *CTLA-4* polymorphisms in cases and controls.**

| Study | Genotype (N) | | | | | | | | Allele frequency (N) | | | | P of HWE |
|---|---|---|---|---|---|---|---|---|---|---|---|---|---|
| | Case | | | | Control | | | | Case | | Control | | |
| | Total | AA | AB | BB | Total | AA | AB | BB | A | B | A | B | |
| +49A>G (rs231775) | | | | | | | | | | | | | |
| Zhifu et al. (2015) | 376 | 174 | 175 | 27 | 366 | 174 | 157 | 35 | 523 | 229 | 505 | 227 | 0.96 |
| Minhas et al. (2014) | 250 | 111 | 113 | 26 | 250 | 105 | 121 | 24 | 335 | 165 | 331 | 169 | 0.20 |
| Li et al. (2012) | 576 | 49 | 281 | 246 | 553 | 54 | 243 | 256 | 379 | 773 | 351 | 755 | 0.74 |
| Sun et al. (2008) | 1,060 | 101 | 485 | 474 | 1,070 | 65 | 446 | 559 | 660 | 1,406 | 576 | 1,564 | 0.15 |
| Sun et al. (2008) | 1,037 | 100 | 455 | 482 | 1,070 | 73 | 451 | 546 | 655 | 1,419 | 597 | 1,543 | 0.12 |
| Wang et al. (2007) | 117 | 48 | 59 | 10 | 148 | 55 | 70 | 23 | 155 | 79 | 180 | 116 | 0.93 |
| Ghaderi et al. (2004) | 197 | 84 | 104 | 9 | 151 | 60 | 72 | 19 | 272 | 122 | 192 | 110 | 0.72 |
| +6230G>A (rs3087243) | | | | | | | | | | | | | |
| Zhifu et al. (2015) | 376 | 257 | 110 | 9 | 366 | 252 | 103 | 11 | 624 | 128 | 607 | 125 | 0.90 |
| Li et al. (2012) | 581 | 361 | 197 | 23 | 566 | 361 | 182 | 23 | 919 | 243 | 904 | 228 | 0.99 |
| Li et al. (2008) | 328 | 32 | 124 | 172 | 327 | 20 | 114 | 193 | 188 | 468 | 154 | 500 | 0.57 |
| Wang et al. (2007) | 117 | 24 | 47 | 46 | 148 | 18 | 56 | 74 | 95 | 139 | 92 | 204 | 0.16 |
| −1722T>C (rs733618) | | | | | | | | | | | | | |
| Zhifu et al. (2015) | 376 | 123 | 186 | 67 | 366 | 137 | 166 | 63 | 432 | 320 | 440 | 292 | 0.30 |
| Li et al. (2012) | 574 | 184 | 276 | 114 | 551 | 207 | 256 | 88 | 644 | 504 | 670 | 432 | 0.55 |
| Li et al. (2008) | 328 | 125 | 163 | 40 | 327 | 111 | 168 | 48 | 413 | 243 | 390 | 264 | 0.22 |
| Erfani et al. (2006) | 282 | 225 | 54 | 3 | 245 | 204 | 41 | 0 | 504 | 60 | 449 | 41 | 0.15 |
| −1661A>G (rs4553808) | | | | | | | | | | | | | |
| Zhifu et al. (2015) | 376 | 273 | 91 | 12 | 366 | 281 | 78 | 7 | 637 | 115 | 640 | 92 | 0.56 |
| Li et al. (2012) | 574 | 405 | 153 | 16 | 551 | 425 | 115 | 11 | 963 | 185 | 965 | 137 | 0.33 |
| Kong (2010) | 315 | 204 | 105 | 6 | 322 | 241 | 76 | 5 | 513 | 117 | 558 | 86 | 0.72 |
| Wang et al. (2007) | 109 | 62 | 45 | 2 | 148 | 111 | 35 | 2 | 169 | 49 | 257 | 39 | 0.68 |
| Erfani et al. (2006) | 282 | 211 | 65 | 6 | 238 | 184 | 43 | 11 | 487 | 77 | 411 | 65 | 0.001 |

**Notes.**

A, the major allele; B, the minor allele; HWE, Hardy-Weinberg equilibrium.

## Meta-analysis results

Seven studies containing 3,613 cases and 3,608 controls focused on breast cancer risk with *CTLA-4* rs231775 polymorphism. As presented in Table 3, significantly decreased risk was observed in the overall population in all the models except heterozygote (G *vs.* A: $OR = 0.86$, 95% CI [0.80–0.92], $P = 0.000$, Fig. 2A; GG *vs.* AA: $OR = 0.68$, 95% CI [0.57–0.81], $P = 0.000$; GG *vs.* AA + AG: $OR = 0.79$, 95% CI [0.71–0.87], $P = 0.000$; AG + GG *vs.* AA: $OR = 0.85$, 95% CI [0.74–0.97], $P = 0.02$;). In subgroup analyses, rs231775 was also found to significantly reduce the breast cancer risk in Chinese and subgroup based on population under allelic, homozygous and recessive models.

There were four studies all of which were from China with 1,402 cases and 1,407 controls investigating the relationship between and breast cancer susceptibility and *CTLA-4* rs3087243 polymorphism. The results presented a significantly lower breast cancer risk in homozygous and recessive genetic models in Chinese women (AA *vs.* GG: $OR = 0.68$,

**Table 3  Meta-analysis results of *CTLA-4* polymorphisms and BC risk.**

| SNP | B *vs.* A | | BB *vs.* AA | | AB *vs.* AA | | BB *vs.* AA + AB | | AB + BB *vs.* AA | |
|---|---|---|---|---|---|---|---|---|---|---|
| | OR (95%CI) | *P* | OR (95%CI) | *P* | OR (95%CI) | *P* | OR (95%CI) | *P* | OR (95%CI) | *P* |
| +49A>G (rs231775) | | | | | | | | | | |
| Overall | 0.86 (0.80–0.92) | **0.000** | 0.68 (0.57–0.81) | **0.000** | 0.92 (0.80–1.06) | 0.23 | 0.79 (0.71–0.87) | **0.000** | 0.85 (0.74–0.97) | **0.02** |
| Chinese | 0.85 (0.79–0.92) | **0.000** | 0.68 (0.56–0.82) | **0.000** | 0.92 (0.73–1.16) | 0.49 | 0.79 (0.71–0.88) | **0.000** | 0.84 (0.65–1.08) | 0.17 |
| PB | 0.86 (0.80–0.93) | **0.000** | 0.70 (0.59–0.84) | **0.000** | 0.91 (0.78–1.05) | 0.19 | 0.80 (0.72–0.89) | **0.000** | 0.85 (0.69–1.04) | 0.12 |
| +6230G>A (rs3087243) | | | | | | | | | | |
| Chinese | 0.87 (0.71–1.07) | 0.20 | 0.68 (0.49–0.95) | **0.02** | 0.99 (0.83–1.19) | 0.94 | 0.77 (0.61–0.97) | **0.02** | 0.87 (0.65–1.17) | 0.36 |
| PB | 0.91 (0.71–1.17) | 0.48 | 0.75 (0.50–1.12) | 0.15 | 1.03 (0.85–1.25) | 0.76 | 0.77 (0.54–1.09) | 0.14 | 1.00 (0.82–1.20) | 0.99 |
| −1722T>C (rs733618) | | | | | | | | | | |
| Overall | 1.09 (0.93–1.29) | 0.29 | 1.15 (0.79–1.68) | 0.47 | 1.13 (0.96–1.32) | 0.15 | 1.11 (0.90–1.37) | 0.32 | 1.14 (0.98–1.33) | 0.09 |
| Chinese | 1.07 (0.88–1.29) | 0.51 | 1.12 (0.77–1.63) | 0.55 | 1.12 (0.94–1.33) | 0.22 | 1.10 (0.89–1.35) | 0.39 | 1.11 (0.86–1.43) | 0.43 |
| PB | 1.19 (1.05–1.34) | **0.007** | 1.37 (1.05–1.78) | **0.02** | 1.22 (1.02–1.47) | **0.03** | 1.21 (0.96–1.54) | 0.11 | 1.26 (1.06–1.50) | **0.01** |
| −1661A>G (rs4553808) | | | | | | | | | | |
| Overall | 1.34 (1.16–1.53) | **0.000** | 1.22 (0.78–1.92) | 0.38 | 1.45 (1.23–1.70) | **0.000** | 1.12 (0.72–1.76) | 0.61 | 1.43 (1.22–1.67) | **0.000** |
| Chinese | 1.41 (1.21–1.63) | **0.000** | 1.59 (0.95–2.67) | 0.08 | 1.47 (1.24–1.75) | **0.000** | 1.45 (0.86–2.43) | 0.16 | 1.48 (1.25–1.75) | **0.000** |
| PB | 1.30 (1.11–1.52) | **0.001** | 1.19 (0.73–1.94) | 0.48 | 1.40 (1.17–1.68) | **0.000** | 1.11 (0.68–1.80) | 0.68 | 1.38 (1.16–1.64) | **0.000** |
| HWE | 1.41 (1.21–1.63) | **0.000** | 1.59 (0.95–2.67) | 0.08 | 1.47 (1.24–1.75) | **0.000** | 1.45 (0.86–2.43) | 0.16 | 1.48 (1.25–1.75) | **0.000** |

**Notes.**

A, the major allele; B, the minor allele; CI, confidence interval; OR, odds ratio; PB, population based; HB, hospital based; SNP, single-nucleotide polymorphism; HWE, subgroup excluding the study departing from HWE.

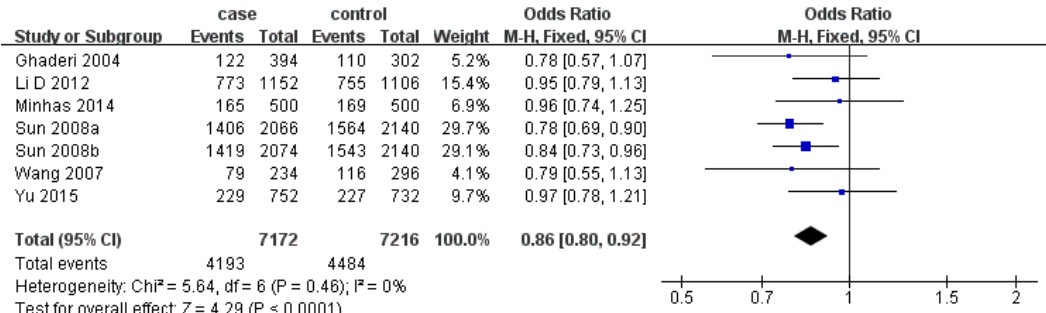

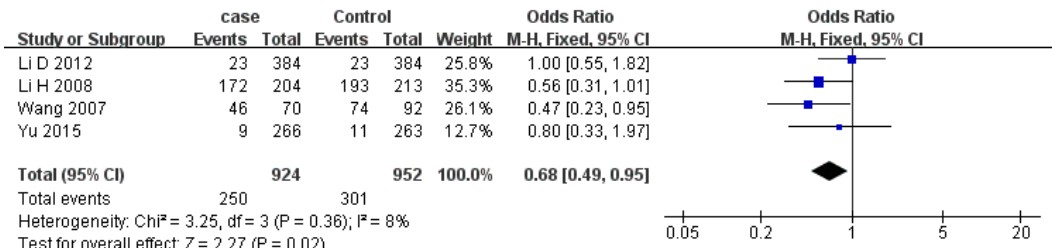

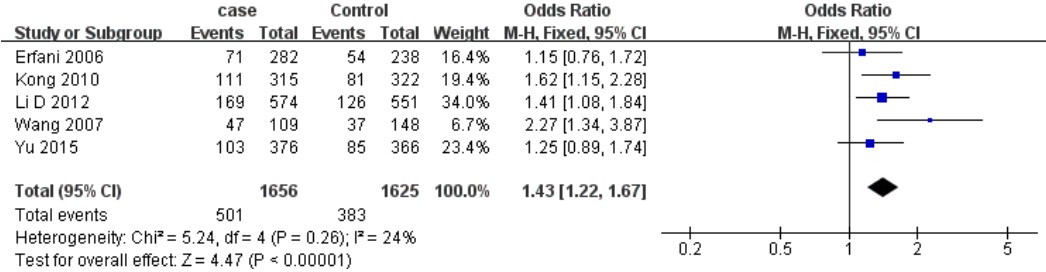

**Figure 2   Forest plots of *CTLA-4* polymorphisms and breast cancer risk.** (A) rs231775 under G *vs.* A; (B) rs3087243) under AA *vs.* GG; (C) rs4553808 under AG + GG *vs.* AA. The squares and horizontal lines correspond to the study-specific OR and 95% CI. The area of the squares reflects the weight (inverse of the variance). The diamond represents the summary OR and 95% CI.

95% CI [0.49–0.95], $P = 0.02$, Fig. 2B; AA *vs.* GG + GA: $OR = 0.77$, 95% CI [0.61–0.97], $P = 0.02$).

For *CTLA-4* rs733618 polymorphism, we aseessed 4 studies containing 1,560 cases and 1,489 controls. Overall, our analysis did not suggest any association between rs733618 and breast cancer susceptibility. However, when stratifying by source of control, rs733618 was observed to increase breast cancer risk based on population in three genetic models (C *vs.* T: $OR = 1.19$, 95% CI [1.05–1.34], $P = 0.007$; CC *vs.* TT: $OR = 1.37$, 95% CI [1.05–1.78], $P = 0.02$; CT *vs.* TT: $OR = 1.22$, 95% CI [1.02–1.47], $P = 0.03$).

Five studies involving 1,656 cases and 1,625 controls investigated the breast cancer risk with *CTLA-4* rs4553808 polymorphism. We observed a higher risk in overall analysis
**Table 4  Heterogeneity-analysis results of *CTLA-4* polymorphisms and BC risk.**

| SNP | B *vs.* A | | | BB *vs.* AA | | | AB *vs.* AA | | | BB *vs.* AA + AB | | | AB + BB *vs.* AA | | |
|---|---|---|---|---|---|---|---|---|---|---|---|---|---|---|---|
| | $I^2$ | *P* | EM | $I^2$ | *P* | EM | $I^2$ | *P* | EM | $I^2$ | *P* | EM | $I^2$ | *P* | EM |
| +49A>G (rs231775) | | | | | | | | | | | | | | | |
| Overall | 0% | 0.46 | F | 45% | 0.09 | F | 29% | 0.21 | F | 27% | 0.22 | F | 41% | 0.12 | F |
| Chinese | 12% | 0.34 | F | 40% | 0.15 | F | 51% | 0.09 | R | 0% | 0.59 | F | 60% | 0.04 | R |
| PB | 6% | 0.38 | F | 39% | 0.14 | F | 39% | 0.15 | F | 0% | 0.56 | F | 51% | 0.07 | R |
| +6230G>A (rs3087243) | | | | | | | | | | | | | | | |
| Chinese | 58% | 0.07 | R | 8% | 0.36 | F | 0% | 0.38 | F | 0% | 0.78 | F | 53% | 0.09 | R |
| PB | 60% | 0.08 | R | 24% | 0.27 | F | 16% | 0.31 | F | 0% | 0.58 | F | 46% | 0.16 | F |
| −1722T>C (rs733618) | | | | | | | | | | | | | | | |
| Overall | 52% | 0.10 | R | 51% | 0.10 | R | 7% | 0.36 | F | 35% | 0.22 | F | 39% | 0.18 | F |
| Chinese | 64% | 0.06 | R | 59% | 0.09 | R | 37% | 0.21 | F | 31% | 0.23 | F | 57% | 0.10 | R |
| PB | 0% | 0.74 | F | 0% | 0.45 | F | 0% | 0.99 | F | 0% | 0.37 | F | 0% | 0.98 | F |
| −1661A>G (rs4553808) | | | | | | | | | | | | | | | |
| Overall | 27% | 0.24 | F | 9% | 0.35 | F | 14% | 0.32 | F | 6% | 0.37 | F | 24% | 0.26 | F |
| Chinese | 0% | 0.48 | F | 0% | 0.99 | F | 32% | 0.22 | F | 0% | 0.98 | F | 24% | 0.27 | F |
| PB | 39% | 0.18 | F | 31% | 0.23 | F | 27% | 0.25 | F | 30% | 0.24 | F | 34% | 0.21 | F |
| HWE | 0% | 0.48 | F | 0% | 0.99 | F | 32% | 0.22 | F | 0% | 0.98 | F | 24% | 0.27 | F |

**Notes.**

EM, Effects model; F, fixed effects model; R, random effects model; PB, population based; HB, hospital based; SNP, single-nucleotide polymorphism; HWE, subgroup excluding the study departing from HWE.

under three models (G *vs.* A: $OR = 1.34$, 95% CI [1.16–1.53], $P = 0.000$; AG *vs.* AA: $OR = 1.45$, 95% CI [1.23–1.70], $P = 0.000$; AG + GG *vs.* AA: $OR = 1.43$, 95% CI [1.22–1.67], $P = 0.000$, Fig. 2C). The results were similar in Chinese subgroup. When stratifying by source of control, rs4553808 was also noted to increase breast cancer risk in allelic and heterozygous models based on population.

### Heterogeneity analysis and sensitivity analysis

As presented in Table 4, no obvious heterogeneity was detected for the four *CTLA-4* polymorphisms in most of the genetic models. For the few in which existed significant heterogeneity ($I^2 > 50\%$), random-effects model was applied.

Each study was sequentially removed to assess the impact of single study on the combined ORs. The result showed that the omission of any study didn't alter the overall estimations substantially, indicating that our meta-analysis results were robust (Fig. 3).

### Publication bias

We implemented Begg's funnel plot and Egger's test to asesess the publication bias. As shown in Fig. 4, funnel plot failed to display obvious asymmetry. The Egger's test result didn't reveal any publication bias for the four SNPs in *CTLA-4* gene and breast cancer risk either (Table 5, $P > 0.05$).

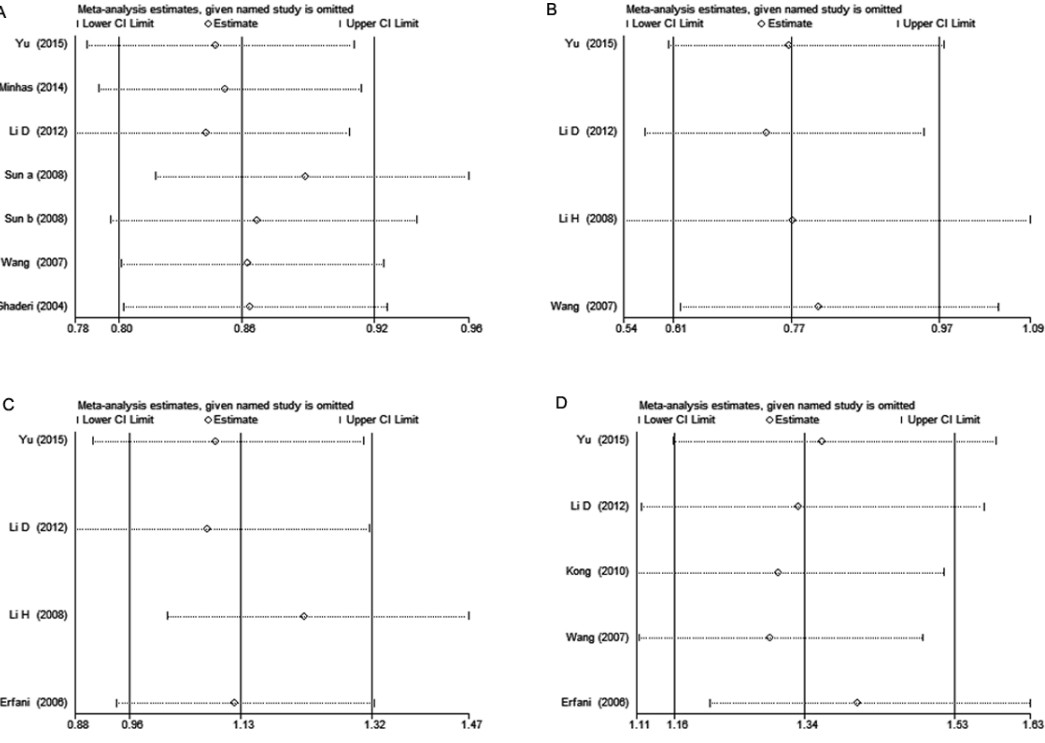

**Figure 3  Sensitivity analysis of *CTLA-4* polymorphisms and breast cancer risk.** (A) rs231775 under G *vs*. A; (B) rs3087243 under AA *vs*. GG + GA; (C) rs733618 under TC *vs*. TT; (D) rs4553808 under G *vs*. A. Each point represents the pooled OR after omitting single study in left column. The two ends of the dotted lines represent the 95% CI.

## DISCUSSION

It was reported that mutation in human *CTLA-4* gene resulted in quantitative reduction of CTLA-4 expression and led to a severe immunoregulatory disorder (*Kuehn et al., 2014*). Several investigations have suggested that particular *CTLA-4* gene polymorphisms are linked to cancer development or progression (*Erfani et al., 2006*; *Tang et al., 2014*; *Wang et al., 2007*). However, the results from those studies remained conflicting. In one previous study, the author found that rs733618 and rs4553808 polymorphisms in *CTLA-4* increased the breast cancer risk whereas rs231775 and rs3087243 polymorphisms did not have significant associations with breast cancer risk (*Li et al., 2012*). However, in other studies, rs3087243 and rs231775 polymorphisms were found to reduce the risk of breast cancer while rs733618 did not associated with breast cancer risk (*Sun et al., 2008*; *Wang et al., 2007*). Since CTLA-4 is important in carcinogenesis and a single study does not have enough statistical power to detect the effects, we carried out this meta-analysis which synthesized the results of the included studies with a statistical analysis of the data from these studies to draw a more reliable conclusion about the association between *CTLA-4* SNPs and breast cancer susceptibility.

In the present meta-analysis, we identified that *CTLA-4* rs231775 had an association with breast cancer susceptibility. We observed a significantly decreased risk in both overall

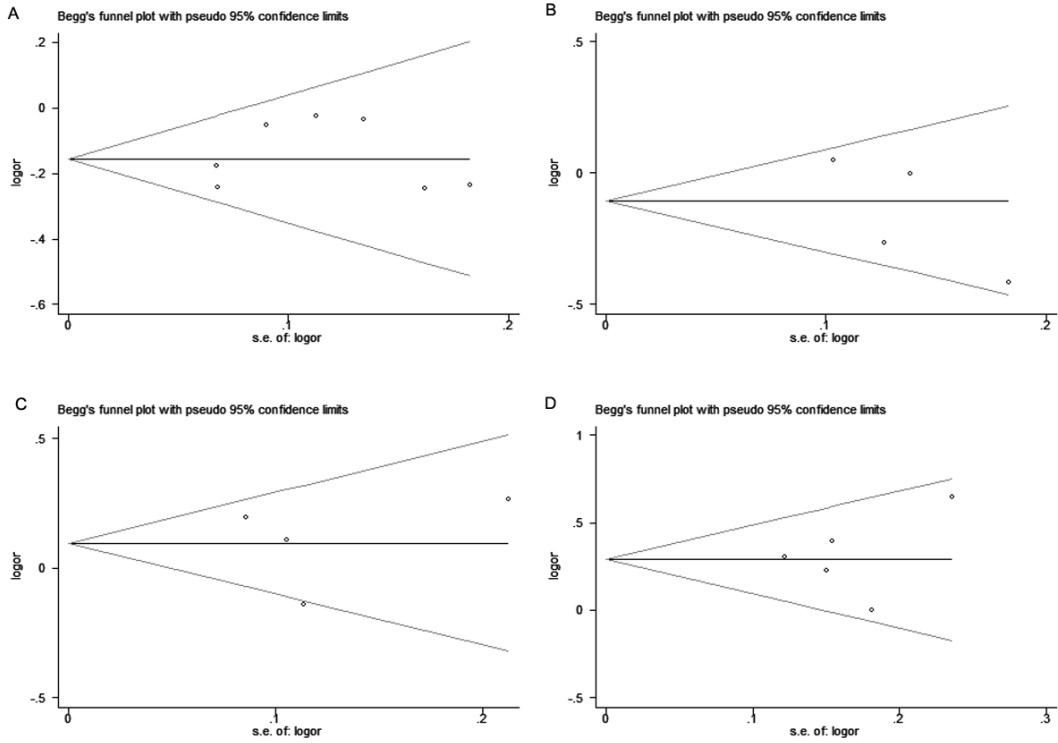

**Figure 4** **Begg's funnel plots of publication bias for the association of *CTLA-4* polymorphisms and breast cancer risk.** (A) rs231775, (B) rs3087243, (C) rs733618, (D) rs4553808 under the allelic model. Each point represents a single study for the indicated association.

**Table 5** **Egger's test result of *CTLA-4* polymorphisms and BC risk based on allele frequency.**

| SNP | Coefficient | SE | t | P | 95% CI |
|---|---|---|---|---|---|
| rs231775 | 0.71 | 1.13 | 0.63 | 0.557 | −2.19–3.61 |
| rs3087243 | −5.19 | 3.20 | −1.62 | 0.246 | −18.94–0.56 |
| rs733618 | −0.03 | 3.17 | −0.01 | 0.993 | −13.67–13.61 |
| rs4553808 | 1.26 | 2.82 | 0.45 | 0.686 | −7.70–10.21 |

**Notes.**
SNP, single-nucleotide polymorphism; 95% CI, 95% confidence interval; SE, standard error.

and subgroup analysis in different genetic models. Some previous meta-analyses have also involved the relationship between rs231775 polymorphism and several tumor sites including breast cancer (*Gao et al., 2014*; *Geng et al., 2014*; *Wang et al., 2015*; *Zhang et al., 2011*). The results suggested the A allele of rs231775 may contribute to breast cancer susceptibility. Our result confirmed that the A allele of *CTLA-4* rs231775 polymorphism has more possibility to increase breast cancer risk than G allele. Nevertheless, our study differs from theirs because we specifically focused on breast cancer and our meta-analysis included more studies than theirs. Therefore, our results are more reliable.

*CTLA-4* rs3087243 polymorphism was found to decrease breast cancer risk under homozygous and recessive genetic models in Chinese. Our results implied that individuals carry AA genotype are less susceptible to breast cancer than those carry GG or (GG + GA)

genotypes. Previous studies also found that rs3087243 was associated with breast cancer susceptibility as a subgroup of several cancer sites (*Yan et al., 2013*; *Zhao, Duan & Gu, 2014*). The results were similar with our research, but our our meta-analysis included one more study and have more statistical power.

We didn't find any relationship between *CTLA-4* rs733618 and breast cancer risk in the overall analysis under any genetic model. However, there was a higher risk in the population-based group under all the genetic models except recessive model. Considering that we selected only four eligible studies and most of them were small-size sample (<500), these results need to be taken with caution. One previous study found a positive signal of rs733618 polymorphism with breast cancer (*Li et al., 2012*) while other two studies showed negative signal (*Erfani et al., 2006*; *Tang et al., 2014*). So further researches with larger sample size should be designed and implemented to validate or refute these conclusions.

In contrast, *CTLA-4* rs4553808 polymorphism was realated to an increased breast cancer risk in both overall and Chinese population in allelic, heterozygous and dominant models. This suggested that *CTLA-4* −1661G allele is more likely to be a risk factor of breast cancer than A allele. Previous meta-analysis also found rs4553808 may increase cancer risk especially for breast cancer (*Geng et al., 2014*; *Yan et al., 2013*; *Zhao, Duan & Gu, 2014*). But these studies investigated the association of this single SNP with various types of cancer while our study specifically focused on breast cancer and investigated several SNPs. Notably, for this SNP, the *P*-value of HWE in control of one study was less than 0.05 (*Erfani et al., 2006*), suggesting that the study population was not representative of a broad population. Nevertheless, we decided to keep this study because deleting it did not affect the pooled ORs significantly.

Several limitations of our research should be noticed. Firstly, the sample size in this meta-analysis was relatively small, especially for rs3087243, rs733618 and rs4553808 polymorphisms. Secondly, our results need to be interpreted with caution since we did not find any studies from Europe, Africa or America and most of the included studies were from China. Therefore, more studies with large population and more ethnic groups are needed to provide sufficient statistical power. Thirdly, other factors such as environmental variants, age, and living habit are generally considered to contribute to cancer susceptibility. Lacking data of these factors for adjustment may impact the estimation of breast cancer risk. Lastly, bias may still exist because we failed to find any studies of other races and we did not have access to gray literature.

## CONCLUSION

In summary, our meta-analysis suggests that rs231775, rs3087243 and rs4553808 polymorphisms in human *CTLA-4* gene significantly associated with breast cancer susceptibility in Asians, particularly in the Chinese population. In consideration of the limitations of our work, further large-scale studies including multi-racial populations are required to confirm our findings.

### Funding

This study was supported by the National Natural Science Foundation, China (No. 81471670); the China Postdoctoral Science Foundation (No. 2014M560791; 2015T81037); the Fundamental Research Funds for the Central Universities, China (No. 2014qngz-04) and the Science and Technology Plan of Innovation Project, Shaanxi province, China (No. 2015KTCL03-06). The funders had no role in study design, data collection and analysis, decision to publish, or preparation of the manuscript.

### Grant Disclosures

The following grant information was disclosed by the authors:
National Natural Science Foundation, China: 81471670.
China Postdoctoral Science Foundation: 2014M560791, 2015T81037.
Fundamental Research Funds for the Central Universities, China: 2014qngz-04.
Science and Technology Plan of Innovation Project, Shaanxi province, China: 2015KTCL03-06.

### Competing Interests

The authors declare there are no competing interests.

### Author Contributions

- Zhiming Dai and Tian Tian performed the experiments, analyzed the data, wrote the paper.
- Meng Wang performed the experiments, analyzed the data, reviewed drafts of the paper.
- Xinghan Liu, Kang Liu and Peng Xu contributed reagents/materials/analysis tools.
- Shuai Lin reviewed drafts of the paper.
- Pengtao Yang, Yi Zheng, Meng Liu and Xuewen Yang prepared figures and/or tables.
- Zhijun Dai conceived and designed the experiments, reviewed drafts of the paper.

### Ethics

The following information was supplied relating to ethical approvals (i.e., approving body and any reference numbers):

The study was approved by the Institutional Review Board of the Second Affiliated Hospital, Xi'an Jiaotong University (Xi'an, China). No. 2016108.

### Data Availability

The raw data is included in Tables 1 and 2.

### Supplemental Information

Supplemental information for this article can be found online at http://dx.doi.org/10.7717/peerj.2815#supplemental-information.

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
