# Peer review of "CTLA-4 polymorphisms associate with breast cancer susceptibility in Asians: a meta-analysis"

_PeerJ, doi:10.7717/peerj.2815_

## Round 0.1 · original submission · Major Revisions

Please attend to all of the reviewer criticisms very thoroughly.

Reviewer 1 ·

Basic reporting

No Comments

Experimental design

No Comments

Validity of the findings

No Comments

Additional comments

In the submission “CTLA-4 polymorphisms associated with breast cancer susceptibility: a meta-analysis” Zhiming Dai et al. evaluated the relationship between four common CTLA-4 polymorphisms and breast cancer risk by a meta-analysis. They showed that rs231775, rs3087243 and rs4553808 polymorphisms in human CTLA-4 gene significantly associated with breast cancer susceptibility, particularly in Chinese population. I suggest the authors revise their manuscript, and show their new findings (especially the results beyond previous studies) explicitly in the revised manuscript.


Specific comments:

1. The authors stated in Introduction section that “Numerous investigations have demonstrated that CTLA-4 genetic polymorphisms may have an association with human breast cancer risk, but the results were in contradiction. (Line 72, Page 5)”. However, they did not show what contradiction in previous studies is. They should explicitly discuss about the contradiction in previous studies, and show what this study contributes to this issue.

2. The authors stated in the Discussion section that “However, the results from those previous studies remain conflicting. (Line 175, Page 10)” The authors should discuss about what conflict in previous studies is, and show what this study contributes to this issue.

3. The authors should explicitly show their new findings, e.g., in the discussion or conclusion section, beyond previous studies. The authors stated in the discussion section that their results “confirmed the previous conclusion that the A allele of CTLA-4 rs231775 polymorphism has more possibility to increase breast cancer risk than G allele. (Line 185, Page 10)”, “the results were identical with our present research. (Line 192, Page 10)”, “In agreement with our result, previous meta-analysis also found rs4553808 may increase cancer risk especially for breast cancer. (Line 205, Page 11)”. However, what new findings of current study are beyond previous results? This new findings should explicitly show in abstract and conclusion section.

4. A few literatures should be cited for sufficient detail & information to replicate. For example, a) Citation for “Preferred Reporting Items for Systematic Reviews and Meta-Analyses (PRISMA) guidelines”; b) Citation for “Newcastle-Ottawa Scale”; Citation for “Woolf’s method”; Citations for “random-effects model” and “fixed-effects model”; Citations for “Begg’s funnel plot and Egger’s test”.

Reviewer 2 ·

Basic reporting

The manuscript titled with “CTLA-4 polymorphisms associated with breast cancer susceptibility: a meta-analysis” studies the relationship between four common CTLA-4 polymorphisms and breast cancer risk by a meta-analysis. The structure of the manuscript is consistent with the journal requirements.The authors argued that the studies on whether particular CTLA-4 gene polymorphisms are linked to cancer development or progression are not consistent(line 175). They should also discuss how they reconcile the inconsistency since their data analysis is based on published literatures.

Experimental design

The choice of methods is fine and the description of methods is good.

Validity of the findings

My main concern is that the authors searched the English and Chinese literatures, and it turn out that in the 10 cases they studied, 7 were from China, 2 were from Iran, and 1 was from India. The authors have to interpret why there is even no single case from Europe, US or South America. Otherwise their samples could be strongly biased, and the conclusions are not solid (line 220-222). It is known that CTLA-4 protein is expressed in activated T cells and plays an essential role in the immune response. Could these be the choice of the key words? Otherwise the research has very limited interest for the general audiences.

Additional comments

First of all the manuscript is poorly written and the abstract need to be rewritten. The authors have to summarize their main findings/conclusions and remove the detailed information from the abstract.
The authors have to improve the English in the manuscript. For example,
Line 3 in the abstract(and line 29): ”but the results remained contradictory.” Contradictory is not the proper word here.
Line 43, change ‘associated’ to associate.
Line 95-96, rewrite this sentence.
Line 113, ‘analyses’ should be analysis.

---

## Round 0.2 · accepted · Accept

Your revision has significantly improved the manuscript.